# Influence of Atherosclerosis-Associated Risk Factors on Expression of Endothelin Receptors in Advanced Atherosclerosis

**DOI:** 10.3390/ijms26052310

**Published:** 2025-03-05

**Authors:** Oliver Herbers, Carsten Höltke, Marco Virgilio Usai, Jana Hochhalter, Moushami Mallik, Moritz Wildgruber, Anne Helfen, Miriam Stölting

**Affiliations:** 1Clinic for Radiology, University Hospital Münster, 48149 Münster, Germany; o.herbers@ludmillenstift.de (O.H.); carsten.hoeltke@ukmuenster.de (C.H.); moushami.mallik@ukmuenster.de (M.M.); anne.helfen@ukmuenster.de (A.H.); 2Department of Vascular Surgery, St. Franziskus Hospital, 48145 Münster, Germany; marcovirgilio.usai@sfh-muenster.de; 3Department of Radiology, University Hospital Munich, LMU Munich, 80336 Munich, Germany; moritz.wildgruber@med.uni-muenchen.de

**Keywords:** atherosclerosis, endothelin receptor, hypertension

## Abstract

Endothelin-1 (ET-1) levels are altered in atherosclerosis, while the roles of the endothelin receptors ET_A_R and ET_B_R during the pathogenesis of atherosclerosis remain unclear. Therefore, the focus of this study was to clarify how endothelin receptors are expressed in advanced human atherosclerotic plaques and how this is related to atherosclerotic risk factors. Ex vivo expression analysis was performed by quantitative real-time PCR (qRT-PCR) of 98 atherosclerotic plaques and controls that were obtained from adult patients undergoing vascular surgery. Correlation analyses of atherosclerosis-promoting factors were accomplished using a linear regression model. We found an overall reduced expression of ET receptors and smooth muscle actin (SMA), a marker of healthy vascular smooth muscle cells, in atherosclerotic plaques, whereas the levels of ET-1 and matrix metalloproteinase 2 (MMP-2), a marker of atherosclerosis progression, remained unchanged. Reduced expression was predominantly correlated with hypertension, which affects both receptors as well as SMA. Age, body mass index (BMI) and gender also correlated with either ET_A_R, ET_B_R or SMA expression in advanced plaques. In contrast, no effect of diabetes mellitus or smoking was found, indicating an ancillary effect of those risk factors. The results of our study indicate that endothelin receptor expression during the pathogenesis of atherosclerosis is predominantly correlated with hypertension.

## 1. Introduction

Cardiovascular diseases, especially ischemic heart disease and stroke, are among the most common causes of death worldwide (World Health Organization). Atherosclerosis is considered to be the primary risk factor [1]. It is defined as a chronic inflammatory disease of medium- and large-sized arteries characterized by the accumulation of lipids, cholesterol and cellular debris within the arterial wall, forming plaques. The pathogenesis of atherosclerosis is initiated by endothelial dysfunction, which is associated with increased permeability and the storage of cholesterol within the subintimal space. Subsequently, inflammatory cells are recruited into the vascular wall, which, due to their high enzymatic activity, can destabilize the atherosclerotic plaque. Accompanying vascular calcification is only the end point of a complex pathomechanism and marks a stage in which the vascular changes have become irreversible [2]. Atherosclerotic plaque development can be depicted by imaging methods like ultrasound, CT or MRI that evaluate carotid intima-media thickness (cIMT) or calcium deposits [3,4], but several studies indicate that the pathogenesis of atherosclerosis may already start during childhood long before clinical manifestations are present [4,5,6]. This highlights the importance of preventing cardiovascular risk factors that trigger endothelial dysfunction and atherosclerosis [6,7,8]. Therefore, treatment often focuses on maintaining a healthy lifestyle to avoid the proinflammatory processes that are believed to start the cascade of atherogenesis [8,9,10,11].

A critical mediator of vascular function is the endothelin system. Endothelin-1, one of three different endothelin isoforms, is the most potent endogenous vasoconstrictor known so far and is thus an important regulator of blood pressure [12,13]. Its effects are mediated through two different G-protein-coupled receptors, the endothelin A receptor (ET_A_R) and the endothelin B receptor (ET_B_R), both typically expressed in different compartments of the vessel wall. While under physiological conditions, ET_A_R is primarily expressed by vascular smooth muscle cells (VSMCs), where binding of ET-1 leads to vasoconstriction, endothelial cells predominantly express ET_B_R, which mediates vasodilatation and ET-1 clearance as well as the initiation of nitric oxide (NO) synthesis [12,13].

ET-1 and its receptors ET_A_R and ET_B_R, however, are also implicated in the development of endothelial dysfunction and atherosclerosis. In atherosclerotic mouse models, ET-1 and ETR expression changes were shown, and an attenuation of plaque formation after treatment of those mice with endothelin receptor antagonists (ERAs) was observed [14,15]. In humans, changes in endothelin-1 and ETR expression as well as a reduction in plaque manifestation due to inhibition of endothelin receptor function have been shown. Winkles et al. found an increased ET-1 expression in atherosclerotic aorta, whereas the ETRs were reduced [16]. Moreover, long-term administration of the ET_A_R antagonist Atrasentan improves coronary endothelial function in patients with endothelial dysfunction and reduces plaque formation [17]. Regarding ET_B_R, an increased expression in human coronary arteries and on inflammatory cells (macrophages, T-lymphocytes) in fibrous plaques and “fatty streaks” was reported [18,19].

Furthermore, implications of ET-1 and its receptors exist in atherosclerosis-triggering diseases affecting lipid metabolism. Hypercholesteremic patients have elevated ET-1 serum and tissue levels, while vascular ET-1 system activity is enhanced in patients with obesity, metabolic syndrome or diabetes mellitus type 2, which has been shown in several studies [20,21,22,23]. This is, in turn, associated with endothelial dysfunction and atherosclerosis progression, leading to diabetic vascular complications [23]. Further pathophysiological conditions favoring the onset or progress of atherosclerosis that are linked to the endothelin system include, e.g., hypertension, diabetic nephropathy or pulmonary arterial hypertension [24,25,26,27,28].

Taken together, despite their unknown precise role in atherogenesis, there is growing evidence that the ETRs are important for atherogenesis. Moreover, there have been treatment options of endothelin-associated diseases involving endothelin receptor antagonists (ERAs) for more than 30 years, as well as several further pharmaceutical approaches, e.g., antibodies targeting endothelin and its receptors are currently under development [12,24,29]. However, there is a need to clarify how both receptors are expressed during atherosclerosis progression and how this expression is influenced by atherosclerosis-associated risk factors that might be treated either by lifestyle modifications or medical treatment, possibly including ERAs.

Our study investigated of the expression of endothelin receptors in advanced atherosclerotic plaques of human origin. Therefore, we analyzed healthy and atherosclerotic human tissue samples ex vivo with respect to expression levels of endothelin and its receptors, as well as SMA and MMP-2, which serve either as a biomarker for ET_A_R-expressing VSMCs or plaque instability. Moreover, we collected patient data concerning atherosclerosis-relevant risk factors and tried to narrow down atherosclerosis-favoring conditions to those that directly affect endothelin receptor expression in advanced atherosclerotic plaques.

## 2. Results

### 2.1. Cohort Data and Pathological State of the Collected Specimen

A total of 255 human samples originating from adult, non-pregnant patients undergoing vascular surgery were collected for further ex vivo processing. Of these, 98 samples showing an RNA integrity value ≥ 5 were used for qRT-PCR analyses. All other specimens were either used for analyses by immunohistochemistry or discarded. The final cohort consisted of 56 atherosclerotic carotid, 16 atherosclerotic femoral and 26 healthy artery specimens (Table 1). All participants exhibited at least one, but mostly multiple, risk factors (see Appendix A).

A fraction of 86% (62 out of 72) of the analyzed atherosclerotic specimens had a North American Symptomatic Carotid Endarterectomy Trial (NASCET) score ≥ 70% or symptomatic atherosclerosis indicated by a diminished walking distance (<200 m) and pain in chest or legs. Both factors indicate advanced atherosclerosis. In summary, 86% of the included patients had advanced atherosclerosis (Table 2).

### 2.2. Histologic Examination of Atherosclerotic Plaques and Arterial Controls

Specimens were examined using immunohistochemistry to evaluate the stage of atherosclerosis (Figure 1). A healthy human artery consists of the typical structure with tunica interna (endothelium), tunica media that contains predominantly smooth muscle cells and the tunica externa (Figure 1A). In contrast, atherosclerotic carotid arteries show a massive thickening of the vessel wall with calcifications at different sites, predominantly within the tunica media restricting the vessel’s lumen (Figure 1B). In an exemplarily depicted femoral specimen, development of atherosclerotic plaque also led to extensive calcification in the media with a high grade of stenosis (Figure 1C). Both atherosclerotic specimens showed typical pathologic characteristics of advanced atherosclerosis, such as lipid deposits and stenosis of the vessel lumen. As anticipated, ET_A_R expression was detected mainly in the tunica media expressed by VSMCs in arterial control specimens (Figure 1D), whereas ET_B_R was depicted predominantly in the tunica interna, expressed by endothelial cells that form the intimal layer (Figure 1E). A robust selective staining of ETRs in atherosclerotic plaques failed, presumably due to the high tissue damage and calcification.

Histologic examination by Elastica van Gieson staining therefore confirmed the advanced stage of the atherosclerotic plaques.

### 2.3. Endothelin Receptor Expression Is Reduced in Advanced Atherosclerotic Plaques

While immunohistochemistry did not reveal ETR expression in the advanced atherosclerotic specimens, the qRT-PCR *ETA-R* (0.377-fold, 95% confidence interval (CI): 0.281–0.393; *p* < 0.001) and *ETB-R* expression (0.557-fold, 95% CI: 0.483–0.631; *p* < 0.001) values were overall reduced in the atherosclerotic specimens when compared to healthy arteries as controls. Concerning tissue *ET1* expression, no differences were found (0.808-fold, 95% CI: 0.719–0.897; *p* = 0.057; Figure 2A).

Interestingly, the expression of smooth muscle actin (*ACTA2*, 0.471-fold, 95% CI: 0.401–0.541; *p* < 0.001), a marker of vascular smooth muscle cells that predominantly express ET_A_R, was also reduced (Figure 2B). This indicates a loss of VSMCs as a consequence of atherosclerotic progression. The expression of *MMP2* (1.061-fold, 95% CI: 0.979–1.143; *p* = 0.536) remained unaffected, although it is—together with other MMPs—a biomarker for altered VSMC behavior and plaque instability during atherosclerotic progression [30,31]. This might be due to the primary occurrence of MMP2 within the extracellular matrix. Comparison of the carotid with femoral specimens showed no significant differences in the expression of either the endothelin system components, *ACTA2* or *MMP2* (Appendix A).

### 2.4. Influence of Atherosclerotic Risk Factors on ET Receptor Expression

Since we observed a reduction in ET receptor expression in the advanced atherosclerotic plaques, we sought to determine whether any risk factors for atherosclerosis correlated with the observed changes. We performed a Kolmogorov–Smirnov test for evaluating Gaussian distribution (Appendix A), followed by linear regression analyses (Figure 3, Appendix A). Factors that were not clearly defined in the questionnaire or where the amount of data was not suitable were excluded. *ETA-R* expression decreased with age (see Figure 3A, Appendix A, *p* < 0.006). Additionally, patients with hypertension exhibited reduced *ETA-R* levels (Figure 3E, Appendix A), *p* < 0.0025), whereas an increased BMI correlated with elevated *ETA-R* expression (Figure 3D, Appendix A), *p* < 0.011). Nevertheless, synergistic reduction due to age and hypertension dominated, as the relative expression levels of *ETA-R* were reduced in qRT-PCR. Interestingly, gender, smoking status and diabetes mellitus did not affect *ETA-R* expression in the analyzed cohort, even though they are considered to be risk factors for atherosclerosis.

*ETB-R* expression was also affected by hypertension (Figure 3E, Appendix A, *p* < 0.002) and in contrast to *ETA-R*, gender additionally correlated with *ETB-R* expression (Figure 3B, Appendix A). Both factors had a deleterious effect on *ETB-R* expression in the advanced atherosclerotic specimens. All other risk factors showed no significant influence on *ETB-R* expression. For *ET1* and *MMP2*, no significant impact of the analyzed risk factors on expression could be detected, but the ACTA2 expression was found to be dependent on age (Figure 3A, Appendix A, *p* < 0.005) and hypertension (Figure 3E, Appendix A, *p* < 0.03), indicating that atherosclerotic plaques in elderly hypertensive people have a lower expression when compared to younger or non-hypertensive people.

Taken together, age, gender, BMI and hypertension were the most prominent factors affecting the expression of endothelin receptors as well as SMA in the advanced atherosclerotic plaques, while nicotine abuse and diabetes mellitus seemed to be subsidiary factors.

## 3. Discussion

The participation of endothelin and its receptors in atherosclerosis development and progression is beyond controversy, but the exact mechanisms and how and when they impact atherosclerosis, especially in humans, are still under consideration. In this ex vivo study, we compared the expression of endothelin-1 and its receptors ET_A_R and ET_B_R in non-atherosclerotic and advanced atherosclerotic human specimens. Parallel analysis of SMA expression served as a biomarker for VSMCs, which predominantly express ET_A_R, whereas MMP-2 served as a biomarker for advanced atherosclerosis. Moreover, we investigated the effect of various modifiable and non-modifiable risk factors of atherosclerosis on the expression of these genes.

Our study shows that the expression of both *ETA*- and *ETB*-receptors was significantly reduced in the advanced human atherosclerotic plaques. This is in accordance with a study by Winkles et al., who reported a reduction in endothelin receptor expression in atherosclerotic plaques of the aorta, while ET-1 levels were increased [16]. Rafnsson et al. also found a decreased expression of ET_A_R in human atherosclerotic plaques of the carotid artery, whereas ET_B_R and ET-1 were upregulated in these samples [32]. A possible reason for the partially opposing results might be that Rafnsson et al. compared atherosclerotic vascular samples from symptomatic with asymptomatic patients rather than atherosclerotic with non-atherosclerotic samples, as presented here. Moreover, we used predominantly advanced atherosclerotic plaques, while most preclinical studies and clinical trials focus on early atherosclerosis.

In one of those trials, treatment with the ET_A_R antagonist Atrasentan led to an improvement in microvascular function in early coronary atherosclerosis, monitored by examination of endothelium-independent coronary flow reserve [33]. Beneficial effects of Atrasentan were also shown in a placebo-controlled group of hypertensive patients with coronary artery disease during a treatment period of 6 months, where no progression of angiographic coronary disease was observed via diagnostic coronary angiography [34]. Moreover, in a preclinical study, ET_A_R antagonism in hyperlipidemic hamsters led to reduced fatty streak development, though the authors suggest an upregulation of ET_A_R in early atherosclerosis [35]. Nevertheless, our data showed a reduced expression of ET_A_R in advanced atherosclerosis, suggesting that molecular changes within atherosclerotic plaques occur during their progression.

We further demonstrated that the expression of *ACTA2*, serving as a biomarker for VSMCs, was reduced in the advanced atherosclerotic plaques. This can be explained by the fact that apoptosis, migration, proliferation and dedifferentiation of smooth muscle cells occurs in advanced atherosclerotic plaques, leading to a decrease in SMA content [36]. Additionally, during atherogenesis, VSMCs show high plasticity and adapt a more macrophage-like phenotype, including a lower expression of SMA [30,36,37]. The reduced expression of *ETA-R* was probably a consequence of the decrease in VSMCs in advanced atherosclerosis, as ET_A_R is primarily expressed by VSMCs. Beyond SMA, we used *MMP2* as a biomarker for advanced atherosclerosis. Interestingly, although advanced atherosclerotic specimens were used, as shown by the NASCET score and immunohistochemistry, no change in *MMP2* expression could be monitored by qRT-PCR. This might be a consequence of the qRT-PCR primer limitations, as MMP-2 typically exists in a preproMMP and proMMP form, which were not targeted by the primers used in our analysis.

In addition to molecular expression analysis, we investigated the influence of modifiable and non-modifiable factors on the expression of the examined markers, which are known to increase the risk of developing atherosclerosis. We found that with increasing age, the expression of both *ETA-R* and *ACTA2* decreased, and that vasodilative *ETB-R* expression was lower in men than in women. Thus, both age and sex, as non-modifiable risk factors, influence endothelin expression. Sex-specific differences in the endothelin system and their role in endothelial function with aging were described before. A reduced ET_B_R expression in aging women has been reported [38] to lead to a decreased ET_B_R-mediated vasodilation, which is attributed to the influence of sex hormones [39]. These findings were supported by estradiol-dependent ET_B_R-mediated vasodilation in young women [40]. Despite this, aging in men leads to an overexpression of endothelin-1 and increased ET-1 mediated vasoconstriction [38]. Hence, the observed reduction in *ETB-R* expression in our analysis is in line with former studies and might partly explain the sex-specific differences.

Concerning the age factor, it must be noted that in the present study, the average patient age between the atherosclerotic and non-atherosclerotic vascular samples differed. Therefore, the observed differences in expression between the two groups need to be interpreted with caution, but due to study limitations, it was not possible to use age-matched groups.

Furthermore, we observed a reduced expression of *ETA-R*, *ET-R* and *ACTA2* in patients suffering from arterial hypertension, reflecting the important pathophysiologic role of the endothelin system in arterial hypertension. Miyagawa et al. described that arterial hypertension leads to increased activation of ET_A_R [41]. Additionally, a high concentration of plasma endothelin-1 in normotensive individuals is related to the development of arterial hypertension [42]. In pulmonary arterial hypertension, where an increased ET-1 expression was shown as well, endothelin receptor antagonists are successfully used in clinical treatment [43,44]. However, our results exhibited a lower expression of *ETA-R* and *ETB-R* as well as *ACTA2* in our patient cohort, where among 69 patients with arterial hypertension, 65 had advanced atherosclerosis.

Interestingly, diabetes mellitus and smoking did not significantly affect the expression patterns in our analyses. Diabetes mellitus is considered to be a major risk factor for atherogenesis. Similar to smoking, the proposed pathomechanism involves increased oxidative stress on the vascular endothelium, resulting in endothelial dysfunction [45,46]. Several studies have shown an increased expression of ET-1 in diabetes mellitus as well as enhanced ET-1-mediated vasoconstriction [47,48,49]. Additionally, Cardillo et al. demonstrated that selective blockade of the endothelin-A receptor in patients with diabetes leads to vasodilation, suggesting an increased activity of the ET-1-ET_A_R pathway [50].

Smoking has also been shown to influence the endothelin system primarily by augmenting ET-1 expression [51,52,53], but an elevated ET_A_R expression is also suggested by studies using ERAs [54].

However, we did not observe a significant impact on the expression of either *ET1* or its receptors. Concerning ET-1, this was predictable due to its function and location in the blood as peptide hormone. As the tissue samples used in the present study only contained a small amount of blood, a possible upregulation of *ET1* was probably not detectable. This applies to all the performed expression analyses using tissue samples and is a clear limiting factor. Concerning the ET receptors, former studies analyzing the effect of smoking used antagonists to draw conclusions concerning receptor expression instead of directly analyzing the expression level. Moreover, our cohort showed a high heterogeneity in terms of age, comorbidities and possibly medication, which may have confounding or neutralizing effects.

We also demonstrated that the expression of *ETA-R* was proportional to BMI. An elevated BMI is thus associated with higher *ETA-R* levels. This finding aligns well with other studies. For instance, in obesity, ET-1-mediated vasoconstriction is increased compared to normal-weight individuals [55]. Another study draws the same conclusion, observing that both overweight and obesity are associated with enhanced ET-1-mediated vasoconstrictor tone [56]. As demonstrated in the current study, this may be attributed to a higher expression of ET_A_R in obesity. Therefore, therapy with ERAs could potentially be beneficial in obesity to reduce the risk of sequelae such as atherosclerosis, especially in young individuals where *ETA-R* expression is higher than in older individuals, as shown in our work.

Some limitations to our study need to be considered. First of all, it was not possible to evaluate sex- or age-matched groups, though there is a risk of misclassification due to selection bias concerning the regression analysis. However, as advanced atherosclerosis naturally occurs in elderly people, there is no way to overcome this limitation if performing a study focusing on advanced atherosclerosis. A comparison of atherosclerotic carotid/femoral specimens with healthy carotid/femoral specimens as a control group would also have been desirable, but this was not possible due to ethical restrictions. Finally, the sample size of the study had an impact on the analyses, as only some, but not all, factors could be used as confounders. To overcome this limitation, a larger sample size will be needed in further studies in the future. Moreover, one could think of a study using in vivo imaging of endothelin receptors to clarify the expression of ETRs during atherosclerosis progression more precisely; however, current imaging probes are for preclinical use only [57,58].

## 4. Materials and Methods

### 4.1. Subject Recruitment and Questionnaire

Patients scheduled for endarterectomy due to lumen-narrowing atherosclerosis or patients scheduled for a vein stripping at the Department of Vascular and Endovascular Surgery at the University Hospital Muenster were recruited for this study to obtain either atherosclerotic carotid or femoral specimens, while lateral femoral artery branches obtained during the vein stripping procedure served as healthy controls. Informed consent was obtained, and each patient filled out a questionnaire concerning risk factors for atherosclerosis. The requested data were age, gender, weight, height, smoking status, pre-existing cardiovascular and renal diseases, arterial hypertension, obesity, family history of cardiovascular diseases, status after stroke and myocardial infarction and ongoing medical treatments (see Appendix A). BMI (body mass index) was calculated as weight (in kg) divided by height (in m) squared. The NASCET score [59], which indicates vascular obliteration, and symptomatic state were collected by the surgeon. Pregnant patients as well as patients younger than 18 years were excluded. This study was approved by the ethical committee of the University of Muenster (protocol number 2016-419-f-S).

### 4.2. Surgical Procedures

Preoperative evaluation of the patients was performed according to local protocols. Surgical procedures were realized under general anesthesia and according to standardized techniques [60,61]. All endarterectomy patients received a periprocedural intravenous heparin bolus dose (5000 I.U.). The explanted carotid or femoral specimens were directly transferred into tubes (Eppendorf, Hamburg, Germany) filled with ice-cold phosphate-buffered saline (PBS). During crossectomy of the vena saphena magna using a standard procedure [61], healthy lateral artery branches of the femoral artery removed at the same time were obtained and similarly transferred into a tube (Eppendorf, Hamburg, Germany) containing ice-cold PBS. All samples were transferred to liquid nitrogen or 4% neutral buffered formalin within half an hour post-resection. In total, 255 samples were obtained for analyses by qRT-PCR or immunohistochemistry.

### 4.3. RNA Isolation

The human tissues were snap-frozen in liquid nitrogen for isolation of total RNA. Prior to RNA isolation, the frozen tissues were homogenized in TRIZOL using a Precellys Evolution homogenizer (Bertin Technologies SAS, Montigny-le-Bretonneux, France) with a user-defined program (see Appendix A). RNA isolation was performed with a Qiagen RNeasy Kit (Qiagen, Hilden, Germany) according to the manufacturer’s instructions. RNA quality was monitored using the Bioanalyzer 2100 (Agilent, St. Clara, CA, USA) of the Core Genomics Unit of the University of Münster. Probes with an RNA integrity value lower than 5 were discarded. Finally, 98 specimens were suitable for qRT-PCR.

### 4.4. Quantitative Real-Time PCR

Quantitative real-time PCR of 98 specimens was performed using a Eppendorf Real Plex Cycler (Eppendorf; Hamburg, Germany) with a KAPA Sybr Fast OneStep qPCR Kit (Sigma Aldrich, Hamburg, Germany) and 20 pg of total RNA per sample according to the manufacturer’s protocol and with Quantitect primer assays (Qiagen, Hilden, Germany; see Appendix A for primer details, Appendix A). Further information on the components of the reaction mixture of qRT-PCR are listed in the attached tables (Appendix A). All samples were run at least two times on different plates containing three technical replicates. GAPDH served as internal control (housekeeper). Analyses of qPCR data were performed using the Relative Expression Software version 2009 2.0.11 (REST, Qiagen, Hilden, Germany) for obtaining expression values.

Data were subsequently processed using Microsoft Excel (Microsoft, Washington, DC, USA) and further processed and analyzed using the SPSS version 29.0.2.0 tool (Statistical Package for Social Science, IBM Corp, Armonk, NY, USA).

### 4.5. Histology and Immunohistochemistry (IHC)

Specimens were fixed in 4% neutral buffered formalin for 24 h and subsequently embedded in paraffin according to standard protocols [62]. Sections of 5 µm thickness were prepared using a microtome (RM2235, Leica; Wetzlar, Germany) and stained for ET_A_R (Abcam ab76259, 1:100; Abcam Limited, Cambridge, UK) or ET_B_R (Proteintech 20964-1-AP, 1:200, ChromoTek GmbH, Planegg-Martinsried, Germany) using a Vectastain kit (PK-6101, Vector Labs; Burlingame, CA, USA) with peroxidase substrate (SK-4100, Vector Labs) or the fluorescent-dye-coupled secondary antibody anti-rabbit Alexa 647 (Invitrogen A32733, 1:200; Thermo Fisher Scientific, MA, USA). Further staining included haematoxylin and eosin as well as Elastica van Gieson (both Morphisto GmbH, Offenbach, Germany), applied according to the manufacturer’s protocol. Negative control sections were carried out without the first antibody. Bright-field images of the sections were scanned using an Eclipse 50i microscope (Nikon, Tokyo, Japan) and documented using the NIS-Elements Br 3.22 software (Nikon, Duesseldorf, Germany). Immunofluorescence images were acquired on an LSM 980 Airyscan 2 confocal microscope using a Plan-Apochromat 20×/0.8 M27 objective and processed using the Zen Blue 3.5 software (Zeiss, Oberkochen, Germany).

### 4.6. Statistical Analysis

Statistical analyses were performed using SPSS version 29.0.2.0 (IBM, Münster, Germany). The relative gene expression values were calculated using the 2^−ΔΔCt^ method [63]. The Kruskal–Wallis test was applied to compare and analyze the differences in expression in the control tissue versus the diseased tissue. Gaussian distribution was proven by a Kolmogorov–Smirnov test (Appendix A). The influence of potential risk factors on the relative gene expression of each protein was examined by a multiple linear regression using age, gender, BMI, smoking, diabetes mellitus and arterial hypertension as covariates. The approximately linear relationship between the non-dichotomous risk factors BMI and age was graphically confirmed to justify the application of linear regression (see Appendix A). A *p*-value below 0.05 was considered statistically significant in all our analyses.

## 5. Conclusions

In summary, our study revealed a reduced expression of both endothelin receptors and SMA in advanced atherosclerotic plaques of human origin. Additionally, *ETA-R* expression negatively correlated with hypertension and age, while it positively correlated with BMI. *ETB-R* expression also negatively correlated with hypertension and gender. Thus, our results provide further insights into the complex molecular pathophysiology of advanced atherosclerosis, especially in humans. They also highlight the importance of hypertension when it comes to endothelin receptor expression. Future preclinical studies and clinical trials should therefore focus on targeted, individualized diagnostics as well as therapies that specifically address the current state of the disease and the expression of potential therapeutic targets.

## Figures and Tables

**Figure 1 ijms-26-02310-f001:**
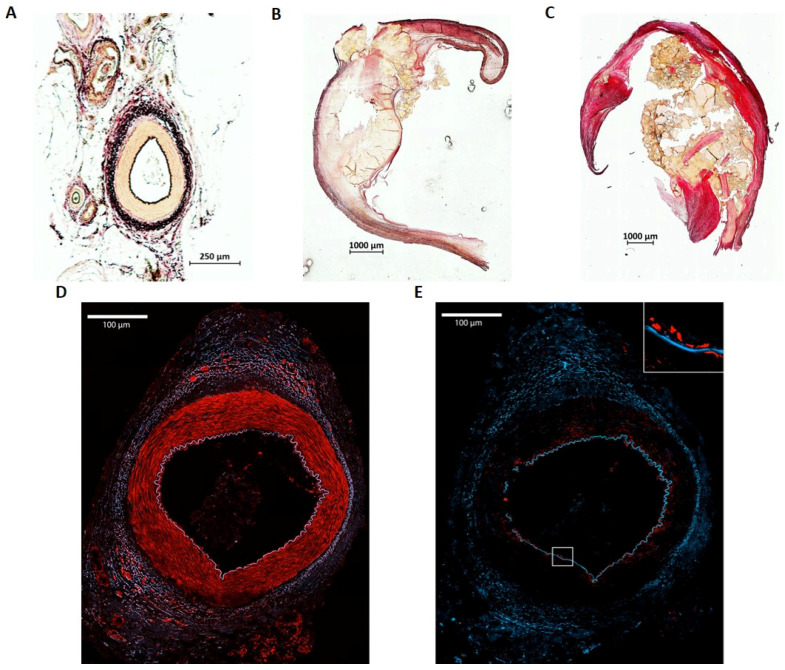
Immunohistochemistry of exemplary tissue specimen. (**A**–**C**) Representative slides after Elastica van Gieson staining of a healthy human artery (**A**) consisting of tunica interna (endothelium), tunica media and tunica externa, (**B**,**C**) advanced atherosclerotic plaques with calcification indicating advanced atherosclerosis. The depicted carotid specimen (**B**) had a NASCET score of 90% and led to symptomatic atherosclerosis. The femoral specimen (**C**) had a NASCET score of 80% and was also symptomatic. (**D**) Lateral femoral artery branch showing ET_A_R expression in the smooth muscle cells of the tunica media. (**E**) Lateral femoral artery branch showing ET_B_R expression in the endothelial layer of the tunica intima. Confocal images show in blue: DAPI, red: ET_A_R/ET_B_R, 20× magnification.

**Figure 2 ijms-26-02310-f002:**
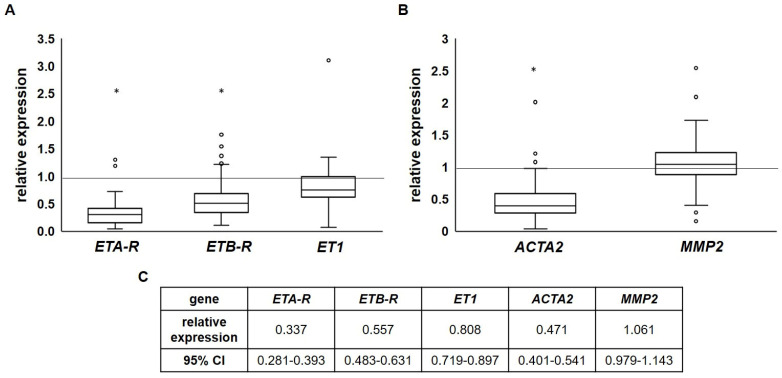
Relative expression of *ET1*, endothelin receptors and markers in atherosclerotic tissue compared to healthy arteries. (**A**) Levels of *ETA-R* and *ETB-R* showed a significant reduction in the atherosclerotic specimen, whereas the expression of *ET1* remained unchanged. (**B**) *ACTA2* expression was also significantly downregulated in the atherosclerotic specimen, while *MMP2* showed no difference in expression when compared to healthy arteries. (**C**) Relative expression and 95% CI for each g (* *p* < 0.05).

**Figure 3 ijms-26-02310-f003:**
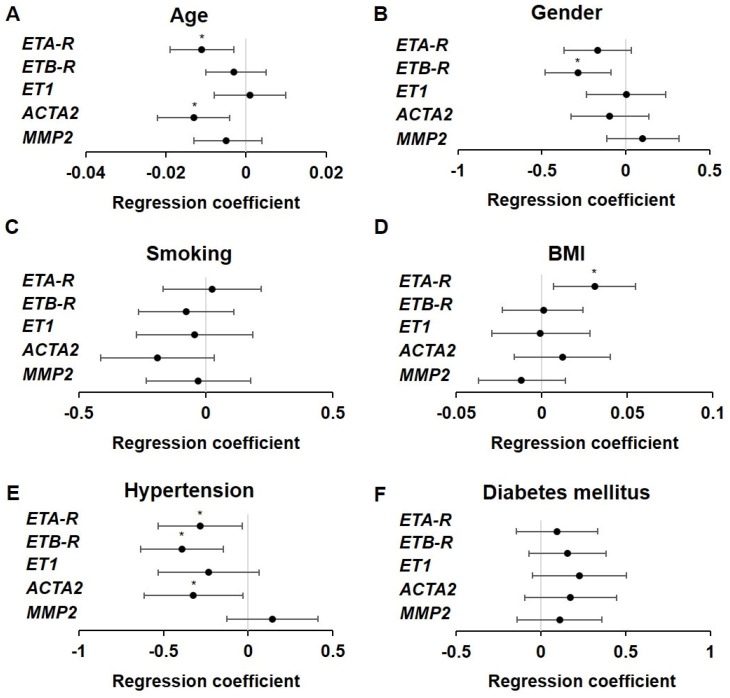
Influence of atherosclerotic risk factors on the relative expression of *ET1*, its receptors and chosen markers. (**A**) The older the individual, the lower the expression of *ACTA2* and *ETA-R*. Age had no significant influence on the expression of *MMP2*, *ET1* and *ETB-R*. (**B**) The expression of *ETB-R* was lower in men than in women. No significant expression changes were found in the other genes. (**C**) Smoking had no significant effect on gene expression. (**D**) The higher the BMI, the higher the expression of *ETA-R*. There was no significant influence on the other genes. (**E**) Hypertension led to a lower expression of *ACTA2*, *ETA-R* and *ETB-R*, while *MMP2* and *ET1* were not changed. (**F**) Diabetes mellitus had no significant effect on gene expression. Regression coefficients and 95% confidence intervals are shown in Appendix A, * *p* < 0.05.

**Table 1 ijms-26-02310-t001:** Characteristics of collected specimens.

Type of Specimen	Healthy Artery *	Atherosclerotic Carotid Artery	Atherosclerotic Femoral Artery	Total *
female	17	17	2	36
male	8	39	14	61
mean age	49	69	70	63

* Gender was not indicated for one artery.

**Table 2 ijms-26-02310-t002:** NASCET score of collected atherosclerotic specimens.

NASCET Score (%)	50	60	70	80	90	w/o Score but Symptomatic	Total
number of specimens	1	3	8	30	9	15	66

## Data Availability

The raw data supporting the conclusions of this article will be made available by the authors on request (except patient-related data).

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
