# Peer review of "Influence of Atherosclerosis-Associated Risk Factors on Expression of Endothelin Receptors in Advanced Atherosclerosis"

_ijms, 2025, doi:10.3390/ijms26052310_

Round 1

Reviewer 1 Report

Comments and Suggestions for Authors

To clarify whether and how proteins of the endothelin axis are expressed in advanced human atherosclerotic plaques and how this is related to atherosclerotic risk factors, this author conducted a cross-sectional study. From this study, this author described as hypertension in conjunction with a dysregulated endothelin system might be a relevant factor during the pathogenesis of atherosclerosis. I thought present study might possesses informative knowledge. However, there are critical problems that should be solved before to be published in scientific journal. Especiily for considering the selection bias is mandatory. All present results could be explained by selection bias that lose scientific values. Therefore, this author should present study design taken into consideration.  

1.      Present description in abstract lacks any important information. Size of present study and specific main results should be clarified even in abstract section.

2.      Without spelling put, many abbreviations are shown even in abstract section.

3.      The characteristics of study population should be described

4.      In introduction section the references that support the following sentences should be shown.

Imaging methods 36 like CT or MRI that evaluate calcium deposits, indicate   that the pathogenesis of athero-37 sclerosis may already start during childhood, long before clinical manifestations are pre-38 sent, highlighting the importance of preventing cardiovascular risk factors.

5In introduction section, this author described the importance of status during childhood. However, present study is a cross-sectional study and the targeted study population is mainly idle aged. And this study evaluated the serum levels of targeted factors on CIMT. Therefore, even this author described about the importance of long turn observational study, this author used the biological factor that enhanced only current condition.

6. How to recruit present study population should be clarified. Since sample size of present study is small, representativeness of present study population should be verified.

7. The methods used in present study is likely to case-controlled study. In this case, healthy control group should be sex and age matched group. However, there is no description about those. Then, in addition to sex distribution, range and distribution of age should be clarified.

8. In discussion section this author described as following

Concerning the factor age, it must be noted, that in the present study the average patient age between the atherosclerotic and non-atherosclerotic vascular samples differed. 240

Therefore, the observed differences in expression between the two groups need to be interpreted with caution. 

   This description explains that there should be significant selection bias that could not adjusted enough. In this case, present results could be explained only by selection bias, even significant correlation were observed. Then, this author should taken into consideration about the risk of misclassification.

9. Specific data should be shown for figures 2 and Figure 4. The specific number of relative expression and regression coefficient should be shown with 95% confidence intervals.

10. Before using linier regression analysis, establishing the linear correlation should be verified, Then, linear correlation should be shown by using plot graphic data.

11. Baseline clinical characteristics of present study population should be clarified. Those data should be shown as tables.

12. The reason why present variables are used as confounder in multiple linear regression analysis should be clarified.

13. What analysis were performed to validate present study model should also clarified.

Reviewer 2 Report

Comments and Suggestions for Authors

I would like to congratulate the authors on their study and manuscript. I think it may be a valuable contribution in the scientific field, especially since much research is directed towards better understanding the pathigenesis of atherosclerosis.

However, I would like to issue the following recommendations before the article is published.

1. The abstract should explain how exactly the study was conducted.

2. The introduction provides ample background, yet a diagram or a figure could better highlight the mechanisms.

3. Also, since there are several research articles already conducted on atherosclerosis, I think the introduction can be further expanded.

4. I think that the paragraph from lines 59-71 would work better in the discussion section.

5. The material and methods should be presented before the results.

6. In the results section please remove the firs paragraph, which is a direction on how to write this section.

7. I would strongly recommend addind a conclusions heading to the summary, in order to better highlight the findings.

Comments on the Quality of English Language

Please revise several minor English language mistakes.

Reviewer 3 Report

Comments and Suggestions for Authors

The article entitled "Influence of atherosclerosis-associated risk factors on expression of endothelin receptors in advanced atherosclerosis" discusses about different types of endothelin receptor the influence of risk factors on them. I would have some suggestions:

1. In the abstract, the authors should define SMA and MMP-2.

2. Which were the exclusion criteria?

3. The authors did not write about the limitations of the study.

4. The article does not have a conclusion section.

5. The authors wrote that they requested many data about antecedents and physical examination. Why they chose just some variables as a risk factor for atherosclerosis? How about other factors? For example waist to hip ratio, family history (were patients with family dyslipidemia included?) etc.

5. Also, were patients with antecedents of MACE included? Is important to divide patients with a prior stroke or acute myocardial infarction from those without those pathologies.

6. The authors should also separate patients with carotid artery and femoral artery atherosclerosis and to compare separately the influence of the risk factors on them. Is there a difference?

7. What about biological investigation? At least CRP and LDL cholesterol.

8. Before the surgical technique, the patients definitely had some different imagistic investigation with different results. What about them and the correlation with endothelin receptors?

9. The most important aspect that is mandatory to also be studied in this manuscript is the treatment. At left hypolipemiant treatment and ACEi for their pleiotropic effects.

Round 2

Reviewer 1 Report

Comments and Suggestions for Authors

According to this author, sex and aged-matched control group is impossible to build in present study. I could understand this author’s claim. However, in this case, sex and aged adjusted model should be taken into consideration. In this case, the effect of sample size on also should be taken into consideration. 

Reviewer 2 Report

Comments and Suggestions for Authors

I believe that all my comments have been properly addressed and the manuscript can ve published in its current form.

Reviewer 3 Report

Comments and Suggestions for Authors

Congratulations for the authors for responded to all of my questions. The article is now suitable for publication.
